# Metagenome-assembled genomes and gene catalog from the chicken gut microbiome aid in deciphering antibiotic resistomes

Yuqing Feng [1], Yanan Wang [2], Baoli Zhu[3], George Fu Gao [3], Yuming Guo[1] & Yongfei Hu [1✉]

Gut microbial reference genomes and gene catalogs are necessary for understanding the chicken gut microbiome. Here, we assembled 12,339 microbial genomes and constructed a gene catalog consisting of ~16.6 million genes by integrating 799 public chicken gut microbiome samples from ten countries. We found that 893 and 38 metagenome-assembled genomes (MAGs) in our dataset were putative novel species and genera, respectively. In the chicken gut, *Lactobacillus aviarius* and *Lactobacillus crispatus* were the most common lactic acid bacteria, and glycoside hydrolases were the most abundant carbohydrate-active enzymes (CAZymes). Antibiotic resistome profiling results indicated that Chinese chicken samples harbored a higher relative abundance but less diversity of antimicrobial resistance genes (ARGs) than European samples. We also proposed the effects of geography and host species on the gut resistome. Our study provides the largest integrated metagenomic dataset from the chicken gut to date and demonstrates its value in exploring chicken gut microbial genes.

[1] State Key Laboratory of Animal Nutrition, College of Animal Science and Technology, China Agricultural University, 100193 Beijing, China. [2] College of Veterinary Medicine, Henan Agricultural University, 450046 Zhengzhou, Henan, China. [3] CAS Key Laboratory of Pathogenic Microbiology and Immunology, Institute of Microbiology, Chinese Academy of Sciences, 100101 Beijing, China. ✉email: huyongfei@cau.edu.cn

Chickens are an important source of meat and eggs for humans, and over 60 billion chickens are estimated to exist worldwide[1]. A large number of microbes, including bacteria and archaea, colonize the chicken gastrointestinal tract and may play vital roles in the degradation of nutrients[2], immune system development[3], pathogen exclusion[4], abdominal fat mass[5], feed efficiency[6], etc. Understanding the roles of the chicken gut microbiome is essential for manipulating gut microbes to promote chicken health and increase the efficiency of chicken production.

In recent years, culture-independent metagenomic approaches have improved our understanding of the diversity, composition, and gene content of gut microbiota in chickens. Similar to other animals, chicken gut flora are dominated by four bacterial phyla, *Firmicutes*, *Bacteroidetes*, *Actinobacteria*, and *Proteobacteria*. Although most microbial genes in the chicken gut are different from those in humans and pigs at the gene sequence level, a large majority of these gene functions are similar among chickens, humans, and pigs[7]. The chicken caeca host the largest number of microbes and play a critical role in chicken gut health, especially due microbial abilities to ferment carbohydrates to produce short-chain fatty acids[8]. As the major enzymes that breakdown plant-derived fibers and degrade dietary carbohydrates and host-derived glycans[9], carbohydrate-active enzymes (CAZymes) have received much attention in gut microbial studies in both chickens and other animals. For example, more than 8000 CAZymes were identified in 155 metagenome-assembled genomes (MAGs) from the chicken gut microbiome[10]. A total of 442,917 genes were predicted to be CAZymes involved in carbohydrate metabolism in 4941 rumen microbial MAGs in cattle[11].

In addition to being involved in nutrient metabolism, the chicken gut microbiome is regarded as a reservoir for antimicrobial resistance genes (ARGs), potentially compromising human health due to the widespread use of antimicrobials in chicken production[12]. We previously profiled regional differences in chicken gut microbial antibiotic resistomes in China[13] and showed that the human gut shares the highest number of mobile ARGs with the chicken gut microbiome[14]. Different mobile genetic elements, such as plasmids, facilitate the spread of antimicrobial resistance among bacteria through horizontal gene transfer (HGT)[15,16]. In both the human and chicken gut microbial communities, HGT-mediated ARG transfer is shaped by the bacterial phylogeny[14].

Although our knowledge of chicken and other animal gut microbiomes was significantly expanded in the high-throughput sequencing era, the detailed functions of gut microbiota in host health and diseases are still difficult to determine. This is partially due to the lack of sufficient reference genomes and genes for gut microbes[17], which impedes the interpretation of sequencing data obtained by culture-independent methods. Currently, reference gene catalogs and/or MAGs of gut microbiomes have been reported for both humans and animals[7,11,18–21]. In chickens, a gene catalog containing ~9 million genes was previously constructed using 495 chicken samples from seven different farms in China[7]. Additionally, we built a gene catalog containing a similar gene number (~8.5 million) using 130 poultry samples collected from live poultry markets in China[13]. For MAGs, 469 draft bacterial genomes were first assembled using the gut metagenomes of 24 chicken samples[21]. Recently, these metagenome-assembled reference genomes were expanded to include 5595 MAGs based on 632 chicken metagenomes[22]. These assembled chicken gut microbial genomes and the gene catalog provide an overview of the chicken gut microbiota landscape. However, along with the increased effort to profile the chicken gut microbiome, expanded and integrated MAGs and gene catalogs are urgently needed.

In the current study, we combined the metagenomic data of the chicken gut microbiome from China and European countries to build an integrated chicken gut microbial reference genomes and gene catalog. We annotated and analyzed the assembled genomes and gene catalog using multiple bioinformatic tools and databases. We also profiled the ARGs in the chicken gut microbiome using the newly assembled MAGs and the gene catalog and compared chicken and human gut antibiotic resistomes. These integrated genomic and gene resources are essential for better understanding the structure and functions of the chicken gut microbiome.

## Results and discussion

**Assembly of 12,339 MAGs from chicken gut microbiome sequencing data.** We assembled expanded MAGs and constructed an integrated gene catalog using metagenomic sequencing data from 799 public chicken gut microbiome samples in China and Europe for the workflow (Fig. 1 and Supplementary Data 1 and 2). After binning the metagenomic contigs, we generated 12,339 dereplicated MAGs (99% average nucleotide identity, ANI) and 1978 dereplicated MAGs (95% ANI) from 19,750 high-quality MAGs (completeness ≥80%, contamination ≤10%, Fig. 2a and Supplementary Data 3). The overwhelming majority of chicken gut microbes were bacteria (1970 genomes), and archaea were extremely scarce (eight genomes). According to the GTDB-Tk assignments, the most dominant phylum was *Firmicutes A* ($n = 822$), followed by *Bacteroidota* ($n = 348$). When redundant MAGs reported in two recent studies[21,22] were removed from our data, a total of 893 species-level MAGs (45.1%) were putative novel species, and a total of 38 genera were candidate novel genera (Supplementary Fig. 1a–c and Supplementary Data 4 and 5). The greatest numbers of novel species-level ($n = 20$) and genus-level ($n = 9$) MAGs belonged to the genera *RC9/Alistipes* and the order *Oscillospirales*, respectively. Two MAGs (MAGs_co_3131 and MAGs_co_10417) could only be assigned to the class level. Strains of *Firmicutes A* and *Bacteroidota* exhibited the highest diversity, as reflected by the Shannon index (Supplementary Fig. 1d), suggesting their contributions to the chicken gut microbiota composition and successful niche occupation and niche/substrate specialization. Strains in the *Bacteroidota* phylum had relatively larger genome sizes and higher proportions of CAZymes (Supplementary Fig. 1e, f), implying their important role in digesting complex carbohydrates; thus, these strains may increase feed efficiency in chickens[23]. Four of the eight archaeal genomes (95% ANI) were novel species, which all belonged to the phylum *Thermoplasmatota* (Supplementary Data 4) and were located relatively close to *Candidatus Methanomethylophilus alvus* Mx-05 in the phylogenetic tree (Supplementary Fig. 2 and Supplementary Data 6). Ca. *M. alvus* Mx-05 was recently isolated from the human gut and demonstrated to have the ability to convert trimethylamine into methane[24].

After mapping reads to the dereplicated MAGs at the strain level, over 85.0% of reads were mapped in most samples (Supplementary Fig. 3a and Supplementary Data 7), demonstrating that our MAGs well represented the chicken gut microbiome. There were 705 samples harboring more than 100 strains, while other samples appeared to contain fewer strains, which may have been due to the limited sequencing depth (Supplementary Fig. 3b). We removed samples with less than 15 million paired reads for most of the downstream analyses, and 477 samples remained (Supplementary Data 8). Variations were observed between samples. At ≥1× coverage, 10,657 MAGs existed in less than 50 gut samples, while only 43 MAGs were present in more than 300 samples (Fig. 2b). Two strains of *Lactobacillus aviarius*

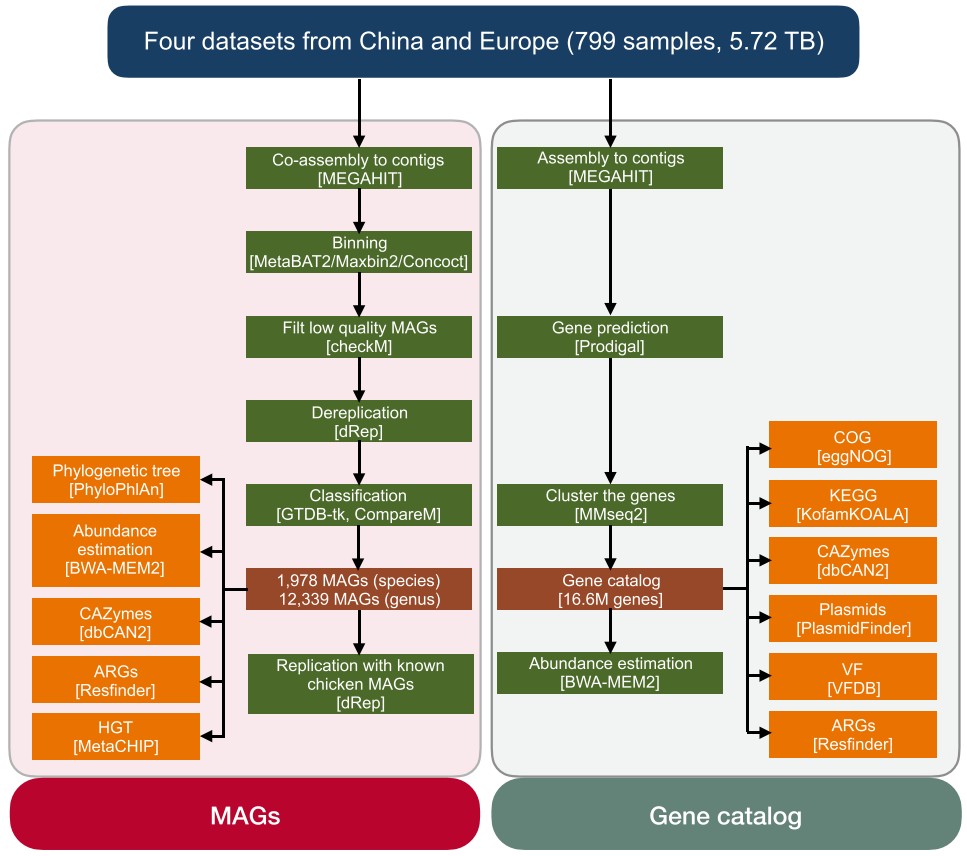

**Fig. 1 Flowchart of the steps and bioinformatic tools applied in assembling, constructing, annotating, and analyzing the reference genomes and microbial gene catalog.** Metagenomic sequencing data from the 799 chicken gut microbiome samples from ten countries were integrated. The MAGs (more than 80% completeness and less than 10% contamination) were clustered to strain-level and species-level genome bins at 99% and 95% ANI, respectively. The phylogenetic tree, CAZymes, ARGs, and HGT of the MAGs were analyzed further. The complete genes were clustered to generate the 16.6-million nonredundant gene catalog. Functions of the genes were annotated to profile CAZymes, virulence factors, and plasmid patterns in the chicken gut microbiome.

and *Lactobacillus crispatus* were very common in the chicken gut microbiota, presenting in over half of the samples (Supplementary Fig. 3c). This implies the importance of these two autochthonous lactic acid bacteria in chickens. A total of 535 MAGs were shared among samples from the ten countries (Supplementary Data 9), most of which belonged to *Limosilactobacillus* ($n = 153$), followed by *Escherichia* ($n = 101$) and *Lactobacillus* ($n = 78$) (Supplementary Fig. 3d and Supplementary Data 10). Among the 535 shared MAGs, 71.2% (381/535) were novel strains, but no novel genera were shared. The novel strains were mainly from the three genera mentioned above: *Limosilactobacillus* ($n = 92$), *Escherichia* ($n = 81$), and *Lactobacillus* ($n = 67$). Due to the strain-specific properties of *Escherichia* and lactic acid bacteria[25,26], strain-level characterization of chicken gut microbes deserves more attention.

**An integrated gene catalog consisting of ~16.6 million genes**. We next built an integrated chicken gut microbial gene catalog (GG-IGC) containing 16.6 million nonredundant genes, which was 1.8 times larger than the previous reference gene catalog of the chicken gut microbiome (CGM-RGC, 9 million)[7]. The lengths of genes in the GG-IGC ranged from 102 bp to 91,812 bp, with a median value of 1083 bp, and more than 63.1% of these genes were complete open reading frames. There were 4,987,193 genes in GG-IGC assigned to 10,665 different KEGG orthologs, compared with 2,611,763 genes assigned to 10,046 KEGG orthologs in CGM-RGC. In addition, 11,290,604 (68.2%) and 6,960,807

(77.0%) genes were annotated with COG functional categories in the GG-IGC and CGM-RGC, respectively (Fig. 2c and Supplementary Data 11). The GG-IGC contained 1.9 and 2.5 times more genes with unknown functions [S] and no-hit results, respectively, than the CGM-RGC (Supplementary Fig. 4 and Supplementary Data 12). These results suggested that the GG-IGC expanded the current chicken gut gene catalog in both microbial gene number and gene function, facilitating better characterization of the roles of the chicken gut microbial community in future multiomics studies. In addition, our results, together with gut microbiome gene catalogs in different hosts from previous studies[18,19,27,28], demonstrated many genes in gut microbial gene catalogs have unknown functions or even lack matches in any database. Therefore, a large number of gut microbes and their functions have not been recognized, which warrants further investigation.

We further annotated the GG-IGC with the databases dbCAN2, virulence factor database (VFDB) and PlasmidFinder to profile CAZymes, virulence factors and plasmid patterns in the chicken gut microbiome. A total of 565,262 CAZyme-encoding genes were annotated in the GG-IGC, corresponding to 371 CAZyme subclasses. The glycoside hydrolase (GH) class was the most abundant in the chicken gut, followed by glycosyltransferase (GT) and carbohydrate-binding module (CBM). The relative proportions of the six CAZyme categories were nearly the same among samples from different countries, but the relative abundance of the CAZyme genes was higher in samples from

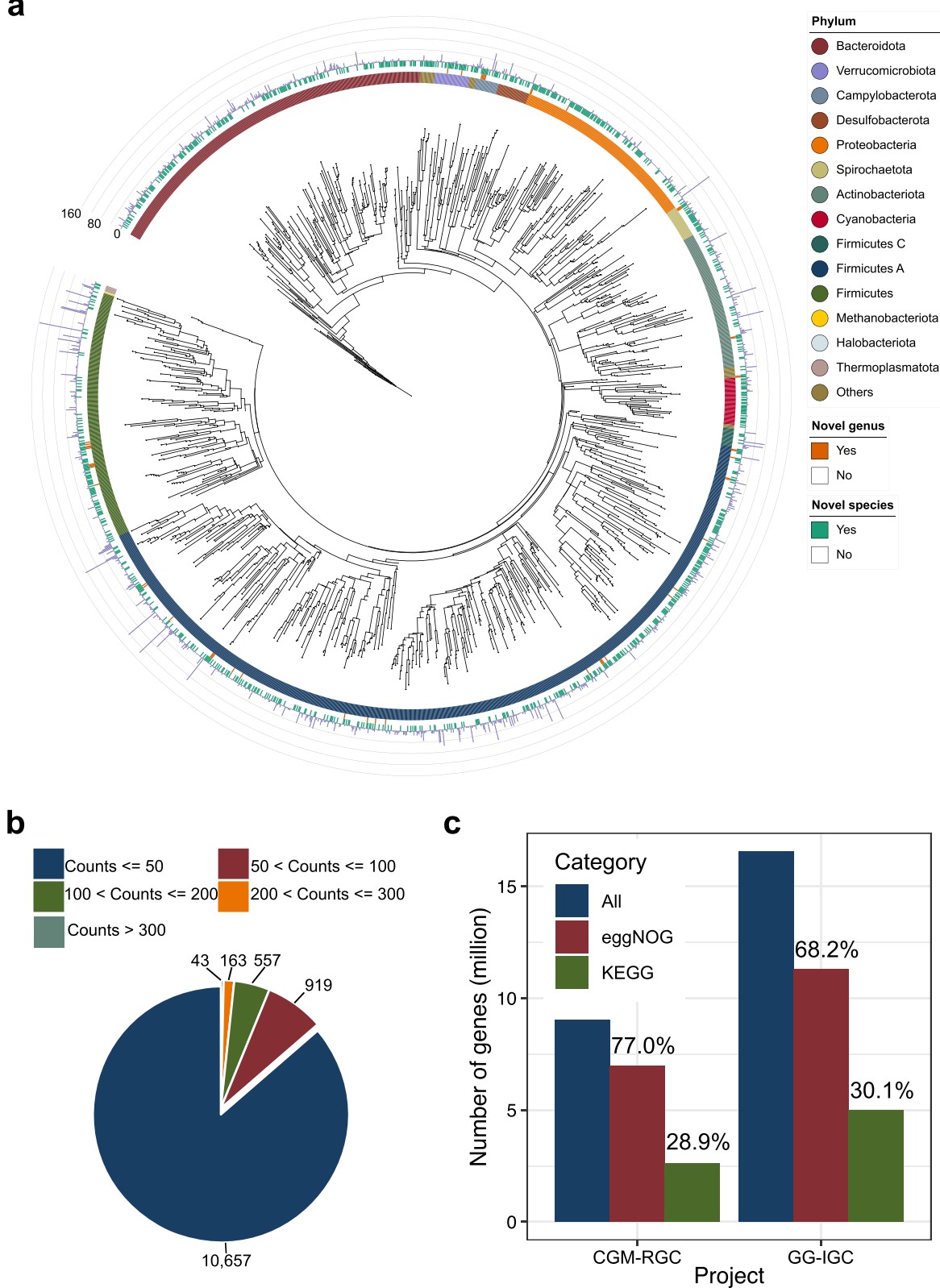

**Fig. 2 Metagenome-assembled 12,339 reference genomes and 16,565,684 nonredundant genes from 799 chicken gut microbiomes. a** Phylogenetic tree of 1970 bacterial and eight archaeal species. The taxonomies of the MAGs were assigned by GTDB-Tk. From the inner to outer rings, the first ring represents the phyla, the second ring represents the novel species (*n* = 893), the third ring represents the novel genera (*n* = 38), and the height of each bar in the fourth ring represents the number of strain-level genomes in each species. **b** Distribution of the 12,339 MAGs among gut samples with the criteria of over 1× coverage. For example, 919 MAGs were present in samples with counts between 50 and 100. **c** Number of genes annotated by eggNOG and KEGG in the CGM-RGC and GG-IGC. The CGM-RGC is a previously published microbial gene catalog of the chicken gut microbiome[7]. The GG-IGC is the integrated microbial gene catalog produced in the present study.

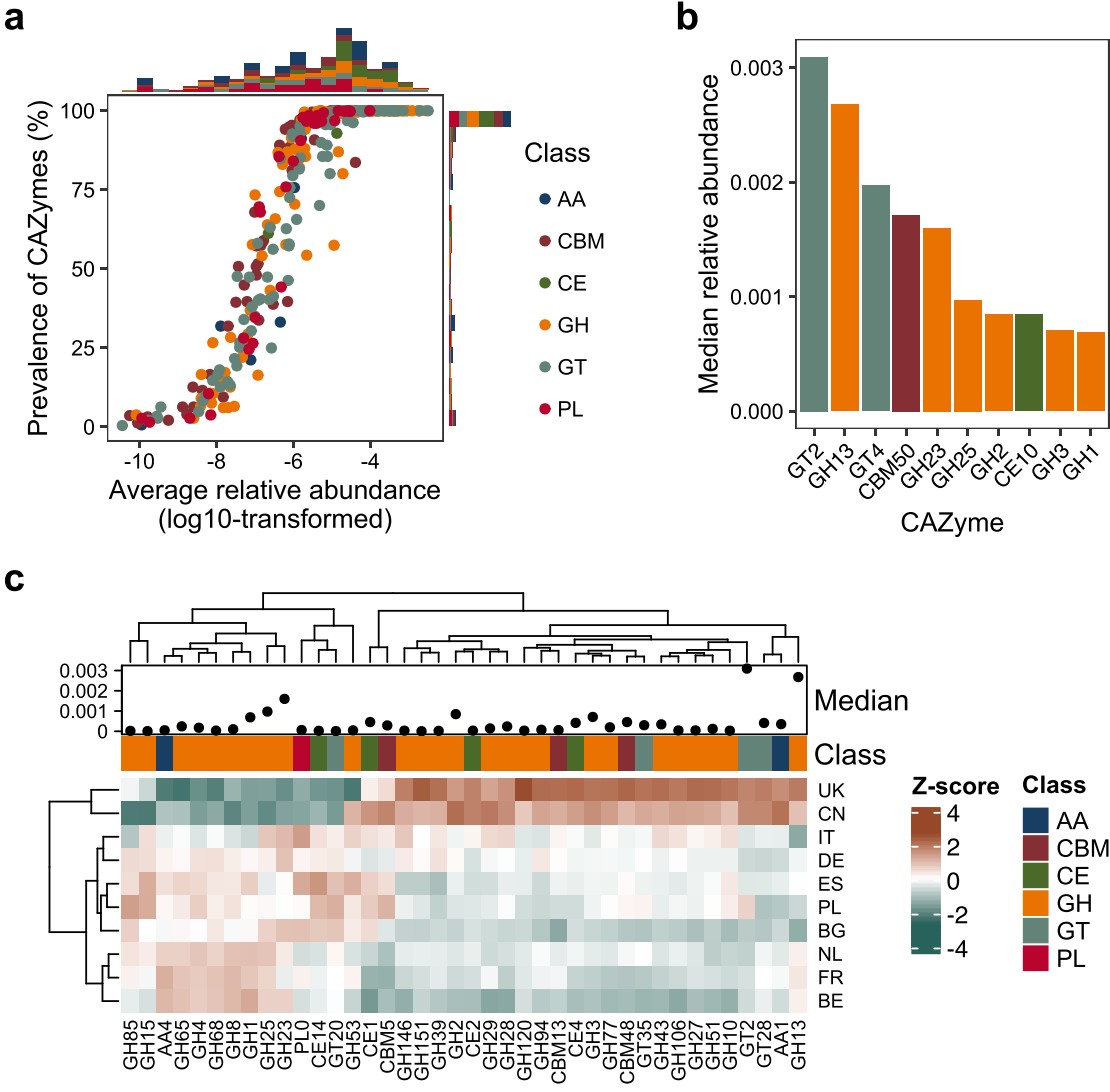

**Fig. 3 Distribution of CAZymes in chicken gut samples. a** Median relative abundance and prevalence of the 371 CAZyme subclasses found in chicken gut samples. **b** The top ten most abundant CAZyme subclasses found in the chicken gut microbiome. The height of each bar is the median value of the relative abundance of each CAZyme subclass. **c** The top 40 CAZyme subclasses with abundance variations among the ten countries. Color: Z-scores of the median values of the relative abundance.

China than in those from European countries ($P < 0.05$, Supplementary Fig. 5). A total of 212 CAZyme subclasses were present in more than 95% of the samples, with average relative abundances ranging from $9.1 \times 10^{-7}$ to $3.2 \times 10^{-3}$ (Fig. 3a), suggesting that most CAZymes were widely distributed among chicken individuals. CAZymes from the GT and GH classes were the most dominant in the chicken gut, with GT2 and GH13 displaying the highest abundance among chickens (Fig. 3b). Additionally, the CAZymes GT2 and GH13 were the top two CAZymes with greatest abundance among samples from different countries. According to the whole CAZyme abundance profile, enzymes with different abundances among samples from the ten countries were mainly in the GH class, such as GH13, GH23, GH25, GH1, GH2, and GH3 (Fig. 3c), and are closely related to cellulose and starch degradation[10]. Notably, GT2, which is a glycosyltransferase, is known to be highly interconnected with other enzymes and may drive temporal changes in the chicken gut microbiota[10]. These results may indicate differential abilities of the chicken gut microbiome in carbohydrate metabolism, probably due to the different chicken lines, raising conditions,

and diet compositions, resulting in different chicken gut microbiota in different countries.

Plasmid typing results showed that 27 types (total 145) of plasmids were present in more than 60% of samples. The top three plasmid types in the chicken gut were repUS43, repUS64 and rep22 (Supplementary Fig. 6a, b and Supplementary Data 13). The commonly present repUS43 may be a carrier of the tetracycline resistance gene *tet(M)* in the chicken gut microbiome, as reported previously[29,30]. Plasmid patterns in samples from China were different from those from the other countries, and they included fewer IncX3, IncX1, p0111, and IncI1 plasmid types (Supplementary Fig. 6c, d). IncX3 plasmids have been described to carry various carbapenemase genes in carbapenemase-producing *Enterobacteriaceae* worldwide[31]. The common VF genes in the chicken gut were carried by bacteria of the *Enterobacteriaceae* family, including *arcB*, *entB*, *entE*, *entF*, *rhs/PAAR*, and *vgrG/tssI* (Supplementary Fig. 7 and Supplementary Data 14), highlighting the pathogenic role of *Enterobacteriaceae*. Plasmid patterns and virulence gene profiles are highly associated with HGT events[32], and more efforts are needed to

accurately identify the mobile genetic elements involved by using metagenomic methods in gut microbiome studies.

**Country-specific chicken gut antibiotic resistomes and comparison with that in humans.** We used the expanded MAGs and the GG-IGC to explore the antibiotic resistome in the chicken gut microbiome. We first analyzed the ARGs in the MAGs and found that 1388 of the 12,339 (11.2%) strain-level MAGs harbored 235 ARG types. The common ARGs in the MAGs were *lnu(C)* ($n = 201$), *mdf(A)* ($n = 120$), and *ant(6)-Ia* ($n = 90$) (Supplementary Fig. 8a and Supplementary Data 15). *Escherichia*, *Romboutsia*, and *Enterococcus* were the top three genera containing the greatest number of ARGs in each genome (≥5 genomes were considered in each genus) (Fig. 4a). HGT prediction analysis indicated that HGT events occurred frequently among *Lachnospiraceae*, *Bacteroidaceae*, *Oscillospiraceae*, *Ruminococcaceae*, and *Acutalibacteraceae* but rarely among *Lactobacillaceae* and other families (Supplementary Fig. 8b). Concerning HGT between gut bacteria, this result reinforces the general notion of lactobacilli as safe. Among the 20,694 genes that may be subjected to horizontal gene transfer, only 17 were ARGs (Supplementary Fig. 9 and Supplementary Data 16). The low positive rate for ARGs in MAGs may have been affected by the plasmid recovery rate and genomic islands during the process of genome binning[33]. Consistent with a previous study, tetracycline and macrolide AMR were most abundant in the chicken gut microbiome at the genome level[34].

We then examined ARGs in our gene catalog. The proportions of each category of ARGs were similar among samples from the ten countries, except for the higher ratio of tetracycline-resistance genes in UK samples (Supplementary Fig. 10a). Chinese samples harbored a higher relative abundance of ARGs than those from Italy, France, the Netherlands, Germany, and the UK ($P < 0.05$), but the ARG diversity in Chinese samples was not high and was even lower than that in samples from Poland, Spain, Italy, and Germany ($P < 0.05$, Fig. 4b, c). The high ARG diversity in samples from these four European countries may be related to the high burden of plasmids in the chicken gut microbiomes in these countries (Supplementary Fig. 6c). Both the abundance and diversity of ARGs were the lowest in the UK samples ($P < 0.05$). Among the 304 ARG types found in GG-IGC, 40 were more abundant in Chinese samples, while 77 were more abundant in European samples (Supplementary Fig. 10b). Chinese samples harbored more ARGs of tetracycline [*tet(W)*, *tet(40)*, *tet(O/W/32/O)*] and aminoglycoside [*aph(3′)-III*, *ant(6)-Ia*, *aac(6′)-aph(2′′)*], but European samples contained more ARGs of macrolide [*lnu(C)*, *lnu(A)*] (Supplementary Fig. 10c). The mobile colistin resistance gene *mcr-1* was found in 5.8% (16/275) of Chinese samples and 7.4% (15/202) of European samples. One Belgian sample was positive for the *mcr-9* gene. By further mapping the original sequencing reads in each sample to all *mcr* gene variants known to date, six *mcr* gene variants, including *mcr-1*, *mcr-3*, *mcr-5*, *mcr-7*, *mcr-9*, and *mcr-10*, were found to be present in the chicken gut microbiome (Supplementary Data 17). The antibiotic resistome in the chicken gut was more related to the plasmid composition than the microbial composition [correlations: 0.201 (microbiota) vs. 0.617 (plasmid), $P < 0.001$, Fig. 4d, e], as confirmed by the residuals between these two comparisons ($P < 0.05$, Fig. 4f).

We then annotated ARGs in the 9.9 million gene catalog of the human gut[35] and compared resistomes between humans and chickens. We showed that 1) the chicken gut contained more ARG types than humans (304 vs. 179), and both human samples and chicken samples from China harbored higher abundances of ARGs than European samples. 2) Gut samples from the same host, either chicken or human, shared a higher number of ARGs (87.8% between Chinese and European chicken samples and 89.4% between Chinese and European human samples). 3) Chinese human and chicken samples shared a slightly higher number of ARGs than that shared between European samples (42.3% vs. 40.8%). 4) Additionally, the effect of geography on the antibiotic resistome was lower than that of the host species (Fig. 4g–i and Supplementary Fig. 11).

## Conclusions

By integrating metagenomic data of the chicken gut microbiome, we expanded the reference microbial genomes in the chicken gut and constructed an integrated gut microbial gene catalog. These data provide a foundation for further functional characterizations and taxonomic assignments of chicken gut microbes. We profiled the antibiotic resistome in the chicken gut using an integrated dataset and revealed the role of plasmids in shaping ARG patterns in the gut microbiome and the host specificity of ARGs in chicken and human gut microbial communities.

## Methods

**Metagenomic data collection.** We collected metagenomic data from four publicly available chicken gut microbiome sequencing projects from China and nine European countries in this study, including PRJEB33338 ($n = 24$)[21], PRJEB22062 ($n = 178$)[34], PRJNA417359 ($n = 495$)[7] and PRJNA408020 ($n = 102$), the latter of which we generated previously[13]. An integrated catalog of 9.9 million reference genes in the human gut microbiome[35] was included for the comparison of gut antibiotic resistomes.

**Metagenome assembly and binning.** Before assembly, low-quality bases (Phred score < 20) and residual Illumina adapter contaminations were excluded using fastp (v0.19.4)[36] and Cutadapt (v1.18)[37], respectively, and reads mapped to chicken, maize, soybean, wheat and zebrafish genomes by BMTagger (v1.1.0) were filtered out. The clean reads of each metagenome were assembled independently using MEGAHIT (v1.1.3)[38]. To increase the generated number of MAGs, coassembly was further performed by dividing all 799 samples into 29 groups using MEGAHIT. The criteria for the grouping of metagenomes were based on various projects and the size of the sequencing data. For metagenomic binning, three methods, i.e., MetaBAT2 (v2.12.1)[39], Maxbin2 (v2.2.6)[40] and Concoct (v1.0.0)[41], were used. A superior bin set from multiple original binning predictions was produced with the Binning_refiner module[42] of MetaWRAP[43]. The completeness and contamination of each bin from the superior bin set were evaluated using CheckM (v1.0.12)[44]. Afterward, bins with ≥80% completeness and ≤10% contamination were retained. To improve the bin quality, bins were reassembled with SPAdes (v3.13.0)[45] in the Reassemble_bins module of MetaWRAP (v1.2.1)[43]. All MAGs were dereplicated at 99% ANI (equivalent to the strain level) and 95% ANI (equivalent to the species level) using dRep (v2.6.2)[46]. GTDB-Tk[47] was used to assign taxonomy to the MAGs. CompareM (v0.1.2, http://github.com/dparks1134/CompareM) was used to calculate the average amino acid identity (AAI) among the MAGs. Genomes were defined as novel strains if the ANI output by GTDB-Tk was <99%. Genomes were determined as novel species if the ANI output by GTDB-Tk was <95%. Genera were defined as novel if all MAGs clustered at 60% AAI were not assigned a genus by GTDB-Tk[47]. We also compared the MAGs with those reported in chicken gut microbiomes in previous studies[21,22] to avoid redundancy. Phylogenetic trees were reconstructed using PhyloPhlAn (v3.0.60)[48]. The phylogenetic tree was based on 400 universal markers defined in PhyloPhlAn and built using the following set of parameters: "-diversity high -fast -remove_fragmentary_entries –subsample fivehundred –min_num_markers 50". The coverage of MAGs at the strain level was calculated as previously described[11]. The standalone run_dbCAN2 (v2.0.11)[49] was used to detect the presence of CAZyme genes in the MAGs. Mass screening of the MAGs for acquired ARGs was performed using ABRicate software (https://github.com/tseemann/abricate), which integrates the Resfinder database[50], VFDB[51], and PlasmidFinder database[52]. All phylogenetic trees were visualized using iTOL (v6.1.1)[53]. Taxonomic classification of the chicken gut microbiome was performed using Kraken 2 (v2.0.9-beta)[54], and we estimated the abundance of each species using Bracken (v2.6)[55].

**Construction of the gene catalog GG-IGC.** Gene prediction of the contigs from each sample was performed by Prodigal (v2.6.3)[56] with the parameter "-p meta". The predicted genes were filtered to remove genes shorter than 100 bp. A non-redundant gene catalog was constructed from the predicted genes by MMseq2[57] with the parameter "easy-cluster -min-seq-id 0.95 -c 0.9" to cluster the genes with the criteria of identity ≥95% and overlap ≥90%. To calculate relative gene abundances, clean reads from each sample were aligned against the gene catalog by BWA-MEM2 (v2.1)[58]. The outputs were converted to the BAM format by

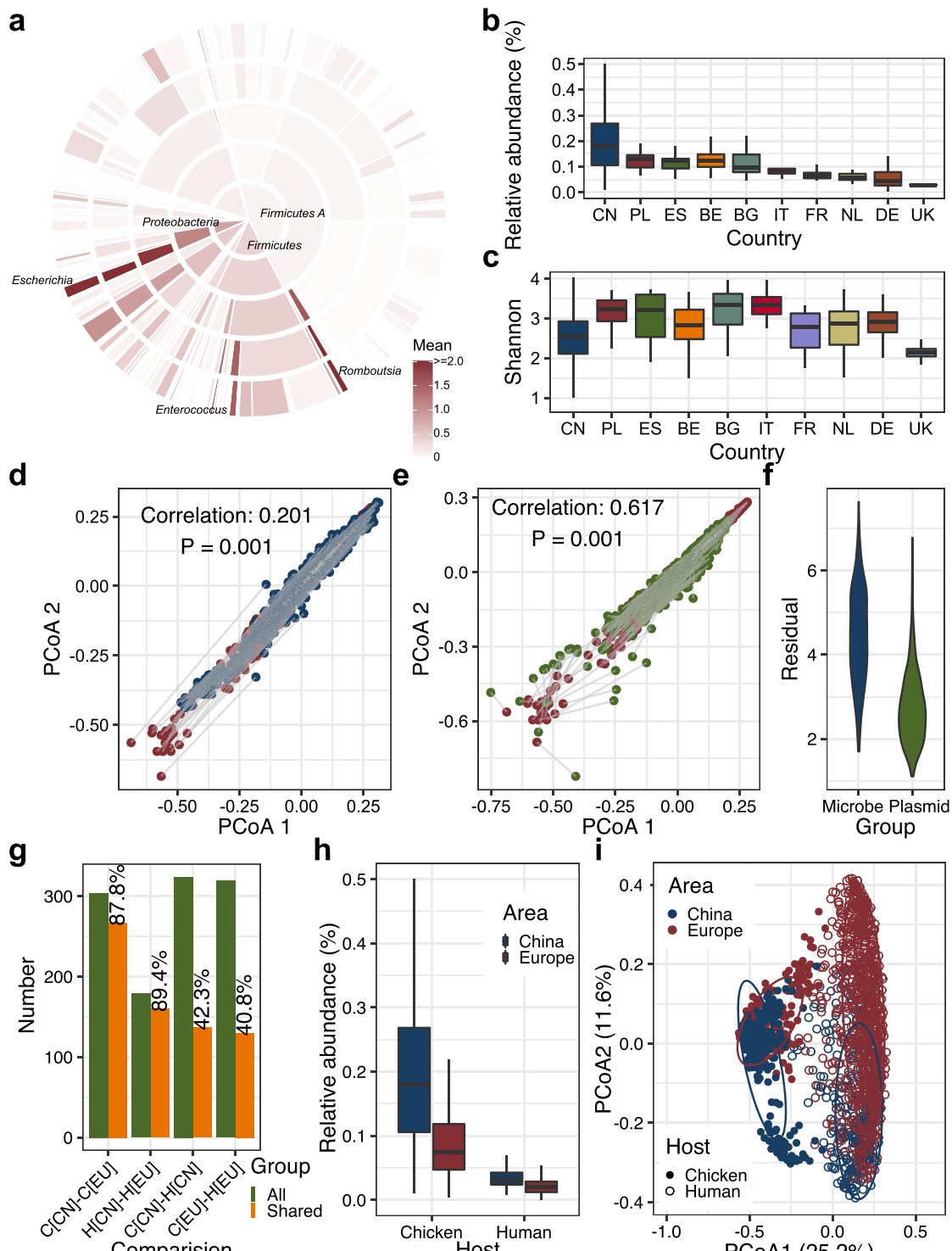

**Fig. 4 Profiling the antibiotic resistome in the chicken gut microbiome. a** Number of ARGs in MAGs at different taxonomic levels. The inner to outer portions represent the phylum level to the genus level. The color represents the mean values of ARGs in specific taxa. **b** Relative abundance of ARGs in samples from different countries. **c** Shannon index of ARGs in samples from different countries. **d** Procrustes analyses of the correlation between the microbial abundances and the ARG abundances. Red dots: ARG abundances; blue dots: microbial abundances. **e** Procrustes analysis of the correlation between the plasmid and ARG abundances. Red dots: ARG abundances; green dots: plasmid abundances. **f** Violin plot of the residuals from the Procrustes analyses in **d**, **e**. **g** Proportion of ARGs shared among microbiomes. C[CN], C[EU], H[CN], and H[EU] denote Chinese chicken samples, European chicken samples, Chinese human samples, and European human samples, respectively. **h** Relative abundance of ARGs in the chicken and human gut microbiomes. **i** Principal coordinate analysis (PCoA) based on the Bray–Curtis distance of the ARG abundance in chickens and humans from China and Europe. CN China, UK United Kingdom, DE Germany, BG Bulgaria, IT Italy, FR France, ES Spain, PL Poland, BE Belgium, NL Netherlands.

SAMtools (v1.11)[59]. Then, the BAM files were translated to abundances using the "jgi_summarize_bam_contig_depth" script from MetaBAT 2[39]. The nonredundant gene catalog was annotated with KofamKOALA (v1.3.0)[60] to assign KEGG Orthology. eggNOG-mapper (v2.0.1b)[61] was used to assign clusters of orthologous groups (COG) functional categories. The presence of CAZyme genes, acquired antimicrobial resistance genes, virulence genes and plasmid replicon genes in the gene catalog was analyzed by the method described above. To explore colistin-resistance genes with low copy numbers that were not included in the gene catalog, clean reads of each sample were aligned to colistin-resistance genes in the ResFinder database[50] using BWA-MEM2[58].

**Identification of horizontal gene transfer**. To identify HGT within chicken gut communities, HGT analysis was performed using MetaCHIP (v1.10.0) on all dereplicated MAGs (clustered with 95% ANI) at the family level. The identification of HGT was performed by the combination of best-match and phylogenetic approaches. The predicted gene flows were visualized using the circlize package in R[62]. The identified HGT genes were further screened against the ResFinder database to identify ARGs using the software ABRicate, as described above.

**Statistics and reproducibility**. The Shannon index and Bray–Curtis distance were calculated by the vegan (v2.5–7) package in R. Differential abundance analysis was performed by a two-tailed Wilcoxon rank sum test. When multiple hypotheses were considered simultaneously, $P$-values were adjusted to control the false discovery rate in R with the method described previously[63]. To determine the effect of the microbiota and the plasmids on the antibiotic resistome, we used Procrustes analysis to determine correlations based on abundance profile Bray–Curtis distances. The correlation between the two datasets was determined by using the "protest" function in R.

**Reporting summary**. Further information on research design is available in the Nature Research Reporting Summary linked to this article.

## Data availability

The gene catalog and MAGs generated in the present study are available in the National Microbiology Data Center (NMDC, https://nmdc.cn/icrggc/) and the Figshare repository with the identifiers "https://doi.org/10.6084/m9.figshare.15982089" and "https://doi.org/10.6084/m9.figshare.15911964")[64,65]. The source data used to create the box plots in the main figures were deposited in Figshare repository ("https://doi.org/10.6084/m9.figshare.16871887")[66]. Any remaining information can be obtained from the corresponding author upon reasonable request.

## Code availability

All R code is available from the corresponding author upon reasonable request. The versions of the software used in the study are described in the "Methods" section.

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

## Acknowledgements

This work was supported by The 2115 Talent Development Program of China Agricultural University and Chinese Universities Scientific Fund. We would like to thank Juncai Ma, Linhuan Wu, and Qinglan Sun from the NMDC team for data preservation and maintenance.

## Author contributions

Y.H. conceived and designed the project. Y.F. analyzed the data and the images in the manuscript. Y.W. collected the metagenomic data. B.Z., G.F.G., and Y.G. provided invaluable feedback and insight for the analysis. Y.F. and Y.H. wrote the initial manuscript. All authors approved the final version of the manuscript.

## Competing interests

The authors declare no competing interests.
