## [Transparent Peer Review File · Communications Biology]

Reviewers' Comments:

Reviewer #1:

Remarks to the Author:

The study presents an integrated chicken gut MAGs and non-redundant genes, as well as some novel findings in ARGs. Though there is no new sequencing data, but the integrated analysis do make contributions to the field of chicken gut metagenomics. In general, the paper is well written and the Figures and Tables are well organized. So, I suggest publication in Communications Biology, except for making minor changes:

(1) The statistics of the assembled contigs, such as contig number, N50 size, total assembly size, must be provided. Completeness rate, contamination rate can't reflect all aspect of MAG, the primary assembly of contigs is more important parameter.

(2) The length and completeness rate of the non-redundant gene set must be provided, the gene set is more useful for later studies unless it is in good continuity.

(3) In the Abstract part, it says "we assembled 12,339 strain-level microbial genomes", I wonder whether the MAGs of current level can be called strain-level microbial genomes, or even species-level microbial genomes. It is better to refer them only as MAG (metagenome assembled genomes), to make difference with the real reference genomes.

(4) Some sentences are hard to understand, and need revision, such as in the Methods/Metagenome assembly and binning:

Bins were dereplicated at 99% and 95% average nucleotide identity (ANI) using dRep (v2.6.2) [32], resulting in each MAG being taxonomically equivalent to a microbial strain and species, respectively.

Reviewer #2:

Remarks to the Author:

The manuscript by Feng et. al entitled "Expanded Metagenome-Assembled Reference Genomes and Microbial Gene Catalog from the Chicken Gut Microbiome Aid in Deciphering Antibiotic Resistome" describes the metagenome assembly and reconstruction of 12,339 strain-level microbial genomes from 799 publicly available metagenomes. Metagenome samples were originated from China and Europe. The authors are reporting that the majority of reconstructed species level genomes are putatively novel species and 75 genera were identified as a putative candidate novel genera. The authors have also constructed a non-redundant metagenome-assembled microbial gene catalog for the chicken gut. The annotation of the genes was executed for the KEGG orthology, COG categories, CAZymes and antibiotic resistance genes profiling. In my opinion, this is an excellent piece of work. However, I have some queries which need to be clarified and resolved in the manuscript.

The abstract is not conclusive about the study, at present it is merely showing the numbers of MAGs and genes. It must reflect the novel results obtained during the comparison of the metagenomes. The major findings and differences between the Chinese and European chicken gut microbiomes.

The introduction part is superficial and it lacks the motivation and aim of the research work. It fails to provide the foundation for the hypothesis. There is a needs to build the background and should be elaborated by justifying the current research work. Include relevant studies and their conclusions related to CAZymes, antibiotic resistance, and horizontal transfer of genes in the chicken gut microbiome.

A recent report published by Gilroy et al 2021 (<https://doi.org/10.7717/peerj.10941>) has used 50 chicken gut metagenomes and publicly available chicken gut metagenomes to reconstruct 5500 bacterial MAGs and 600 bacteriophage genomes. I suggest comparing the novel MAGs reported by Gilroy et al to avoid the redundancy in reporting the novel MAGs obtained during the present work.

Similar to the previous comment, Glendinning et. al. (2020) paper has reported 460 novel strains, 283 novel species, and 42 novel genera. I wonder if the same MAGs were reconstructed in this research work. The authors should discuss the reconstruction of MAGs in comparison to the novel MAGs obtained by Glendinning et al. and other previous reports.

Table S7 and S8: How many shared genera were novel?

Line 97: What does the CGM-RGC stand for. It is confusing for the readers to understand the abbreviation. You need to properly define the abbreviations in the manuscript.

Line no. 103-104: Why the genes were not showing the hits against the database. What are the reasons for higher unannotated genes in GG-IGC? There might be a database limitation for the lower number of annotated proteins. The authors should discuss the consideration/limitation of the techniques in the text.

Line no. 198-200: As far as I understand that each metagenome was assembled independently. It needs to be written in the methods. To improve the quality of binning a co-assembly was performed, what criteria were used for the grouping of metagenomes before the co-assembly?

Some minor comments:

Line no. 42: What are the other roles of chicken gut microbiota? Please mention them.

The data is available on the website <https://nmdc.cn/icrggc/>. However, It will be more convenient to access the MAGs data from the public database. Reconstructed MAGs must be submitted to the public genome database.

Line no. 195: What number was considered as low-quality for the nucleotides?

Line no. 219: Remove "and so on" from the sentence.

Reviewer #3:

Remarks to the Author:

This work includes genomes recovery and data analysis of large number of chicken gut metagenomic datasets which significantly expanding the reference genomes available from the chicken gut microbial communities, cataloging the genes prevalent in the gut systems and assessing the extent of HGTs and their effects on the spread of the antibiotic resistance.

The number and size of datasets analyzed as well as the number of reference genomes recovered in this study make this study quite interesting and qualify it to be an important contribution in the field of animal associated microbiome research in general.

Major takes on this work: -

- 1- This study is more like data dump than actual analysis paper, it lacks insights, clear conclusions or future recommendations on how to use this massive amount of the data to advance the field.
- 2- The title and the abstract are bit misleading, they gave me an impression that this study based on datasets created by the group, however it turned to be a re-analysis of publicly available datasets.
- 3- The analyses performed on the MAGs were shallow at all levels and I failed to see any conclusive results. The results were more like describing computational tools outputs without spending time or effort to put the results into any context.

Other points: -

- 1- Figure 1a, reorder the labels to match the order of the phyla organization on the tree.

2- Figure 2b looks like a cartoon than an actual tree with no references to see the position of the taxa, please redo the archaea tree with the addition of references, bootstrap needs to be presented and the phylogenetic trees in newick format should be attached as supplementary documents.

3- Ln 75: Having high diversity doesn't reflect abundance. This statement is not quite accurate. It only reflects a successful niche occupation and probably niche/substrate specializations.

4- Figure S2 need to be normalized to the genome size and data needs to be represented by number of genes/Mbps to reflect the real saccharolytic capabilities of the recovered genomes.

5- Ln 107: Some qualitative analysis of the CAZyme classes present is required to get a deeper understanding of the role of CAZymes in niche occupation by the inhabiting microbes.

Responses to the reviewers' comments

Reviewers' comments:

Reviewer #1 (Remarks to the Author):

The study presents an integrated chicken gut MAGs and non-redundant genes, as well as some novel findings in ARGs. Though there is no new sequencing data, but the integrated analysis do make contributions to the field of chicken gut metagenomics. In general, the paper is well written and the Figures and Tables are well organized. So, I suggest publication in Communications Biology, except for making minor changes:

Reply: We appreciate these encouraging comments.

(1) The statistics of the assembled contigs, such as contig number, N50 size, total assembly size, must be provided. Completeness rate, contamination rate can't reflect all aspect of MAG, the primary assembly of contigs is more important parameter.

Reply: We added a table in the Supplementary Materials (Supplementary Table 2) displaying the statistics of the assembled contigs, as suggested. We also added the N50 size in Supplementary Table 3. (Lines 99-101)

(2) The length and completeness rate of the non-redundant gene set must be provided, the gene set is more useful for later studies unless it is in good continuity.

Reply: We thank you for this suggestion. We added the missing information in the revised manuscript as follows: "The lengths of genes in the GG-IGC ranged from 102 bp to 91,812 bp, with a median value of 1,083 bp, and more than 63.1% of these genes were complete open reading frames." (Lines 140-141)

(3) In the Abstract part, it says "we assembled 12,339 strain-level microbial genomes", I wonder whether the MAGs of current level can be called strain-level microbial genomes, or even species-level microbial genomes. It is better to refer them only as MAG (metagenome assembled genomes), to make difference with the real reference genomes.

Reply: We deleted "strain-level" from the Abstract section and revised the text to "Here, we assembled 12,339 microbial genomes and constructed a gene catalog consisting of ~16.6 million genes by integrating 799 public chicken gut microbiome samples from ten countries." (Lines 33-34)

(4) Some sentences are hard to understand, and need revision, such as in the Methods/Metagenome assembly and binning:

Bins were dereplicated at 99% and 95% average nucleotide identity (ANI) using dRep (v2.6.2) [32], resulting in each MAG being taxonomically equivalent to a microbial strain and species, respectively.

Reply: We revised the sentence to “All MAGs were dereplicated at 99% ANI (equivalent to the strain level) and 95% ANI (equivalent to the species level) using dRep (v2.6.2).” (Lines 258-259)

Reviewer #2 (Remarks to the Author):

The manuscript by Feng et. al entitled “Expanded Metagenome-Assembled Reference Genomes and Microbial Gene Catalog from the Chicken Gut Microbiome Aid in Deciphering Antibiotic Resistome” describes the metagenome assembly and reconstruction of 12,339 strain-level microbial genomes from 799 publicly available metagenomes. Metagenome samples were originated from China and Europe. The authors are reporting that the majority of reconstructed species level genomes are putatively novel species and 75 genera were identified as a putative candidate novel genera. The authors have also constructed a non-redundant metagenome-assembled microbial gene catalog for the chicken gut. The annotation of the genes was executed for the KEGG orthology, COG categories, CAZymes and antibiotic resistance genes profiling. In my opinion, this is an excellent piece of work. However, I have some queries which need to be clarified and resolved in the manuscript.

Reply: We are grateful for your positive comments.

The abstract is not conclusive about the study, at present it is merely showing the numbers of MAGs and genes. It must reflect the novel results obtained during the comparison of the metagenomes. The major findings and differences between the Chinese and European chicken gut microbiomes.

Reply: Many thanks for these suggestions. We have revised the Abstract (limited to 150 words) as follows: “Gut microbial reference genomes and gene catalogs are necessary for understanding the chicken gut microbiome. Here, we assembled 12,339 microbial genomes and constructed a gene catalog consisting of ~16.6 million genes by integrating 799 public chicken gut microbiome samples from ten countries. We found that 893 and 38 metagenome-assembled genomes in our dataset were putative novel species and genera, respectively. In the

chicken gut, *Lactobacillus aviarius* and *Lactobacillus crispatus* were the most common lactic acid bacteria, and glycoside hydrolases were the most abundant CAZymes. Antibiotic resistome profiling results indicated that Chinese chicken samples harbored a higher relative abundance but less diversity of antimicrobial resistance genes (ARGs) than European samples. We also proposed the effects of geography and host species on the gut resistome. Our study provides the largest integrated metagenomic dataset from the chicken gut to date and demonstrates its value in exploring chicken gut microbial genes, e.g., ARGs.” (Lines 32-41)

The introduction part is superficial and it lacks the motivation and aim of the research work. It fails to provide the foundation for the hypothesis. There is a needs to build the background and should be elaborated by justifying the current research work. Include relevant studies and their conclusions related to CAZymes, antibiotic resistance, and horizontal transfer of genes in the chicken gut microbiome.

Reply: We agree with these comments. We carefully revised the Introduction section to include CAZymes, antibiotic resistance, and horizontal transfer. (Lines 47-93)

A recent report published by Gilroy et al 2021 (<https://doi.org/10.7717/peerj.10941>) has used 50 chicken gut metagenomes and publicly available chicken gut metagenomes to reconstruct 5500 bacterial MAGs and 600 bacteriophage genomes. I suggest comparing the novel MAGs reported by Gilroy et al to avoid the redundancy in reporting the novel MAGs obtained during the present work.

Reply: Many thanks for this great suggestion. We compared the novel MAGs in the present study with those in previous publications^{1,2} and removed the redundancy in the present study. The results included 9,845 novel strains, 893 novel species and 38 novel genera, decreased from 11,783, 1,449 and 75, respectively. We modified the data in the manuscript and Supplementary Materials accordingly. (Lines 104-106 and 263-264)

Similar to the previous comment, Glendinning et. al. (2020) paper has reported 460 novel strains, 283 novel species, and 42 novel genera. I wonder if the same MAGs were reconstructed in this research work. The authors should discuss the reconstruction of MAGs in comparison to the novel MAGs obtained by Glendinning et al. and other previous reports.

Reply: We agree with this comment. We compared all the MAGs in this study with those in two other papers^{1,2} and found that some of the novel MAGs that we previously showed have been reported. We have modified the related results, as explained in the previous reply.

Table S7 and S8: How many shared genera were novel?

Reply: Among the 535 shared MAGs, 71.2% (381/535) were novel strains, but no novel genera were shared. We explained this point in the text. (Lines 131-132)

Line 97: What does the CGM-RGC stand for. It is confusing for the readers to understand the abbreviation. You need to properly define the abbreviations in the manuscript.

Reply: The CGM-RGC was defined in a previous study and is a reference gene catalog of the chicken gut microbiome³. We added the full name to its first appearance in the manuscript. (Lines 139-140)

Line no. 103-104: Why the genes were not showing the hits against the database. What are the reasons for higher unannotated genes in GG-IGC? There might be a database limitation for the lower number of annotated proteins. The authors should discuss the consideration/limitation of the techniques in the text.

Reply: Thank you for providing this interesting question. Actually, the annotation rates of gene catalogs generated from gut microbiomes are usually low, as can be found in other reports investigating gut gene catalogs in different hosts. Chen *et al.* reported that 61.5% and 16.6% of the genes in the pig gut microbiome were annotated to COG functional categories and KEGG orthologous groups (KOs), respectively⁴. Xie *et al.* reported that 65.0% and 32.9% of the genes in the ruminant gut microbiome were annotated to COG functional categories and KOs, respectively⁵. Approximately 80% of the genes in the human gut metagenomes were annotated to COG functional categories⁶. Almeida *et al.* reported that 41.5% of genes in their gene catalog of the human gut microbiome could not be annotated⁷. This may be because many gut microbes and the genes that they carry have not been recognized. Further studies are highly needed to explore the functions of these unannotated genes in gut microbiomes. We added a related discussion in the text. (Lines 148-152)

Line no. 198-200: As far as I understand that each metagenome was assembled independently. It needs to be written in the methods. To improve the quality of binning a co-assembly was performed, what criteria were used for the grouping of metagenomes before the co-assembly?

Reply: Thank you for your suggestion. We revised the text to “The clean reads of each metagenome were assembled independently using MEGAHIT (v1.1.3). To increase the generated number of MAGs, coassembly was further performed by dividing all 799 samples

into 29 groups using MEGAHIT. The criteria for the grouping of metagenomes were based on various projects and the size of the sequencing data.” (Lines 249-252)

Some minor comments:

Line no. 42: What are the other roles of chicken gut microbiota? Please mention them.

Reply: The text has been revised accordingly, as follows: “A large number of microbes, including bacteria and archaea, colonize the chicken gastrointestinal tract and may play vital roles in the degradation of nutrients, immune system development, pathogen exclusion, abdominal fat mass, feed efficiency, etc.” (Lines 48-50)

The data is available on the website <https://nmdc.cn/icrggc/>. However, It will be more convenient to access the MAGs data from the public database. Reconstructed MAGs must be submitted to the public genome database.

Reply: We deposited the gene catalog and MAGs into Figshare (10.6084/m9.figshare.15982089 and 10.6084/m9.figshare.15911964) according to the instructions of Communications Biology. Related information was added to the Data availability section. (Lines 309-311)

Line no. 195: What number was considered as low-quality for the nucleotides?

Reply: A Phred score of nucleotides under 20 was considered to indicate low quality. We trimmed the nucleotides using this Phred score. We revised the sentence to “Before assembly, low-quality bases (Phred score < 20) and residual Illumina adapter contaminations were excluded using fastp (v0.19.4) and Cutadapt (v1.18), respectively, and reads mapped to chicken, maize, soybean, wheat and zebrafish genomes by BMTagger (v1.1.0) were filtered out.” (Lines 247-249)

Line no. 219: Remove “and so on” from the sentence.

Reply: The text has been revised accordingly.

Reviewer #3 (Remarks to the Author):

This work includes genomes recovery and data analysis of large number of chicken gut metagenomic datasets which significantly expanding the reference genomes available from the

chicken gut microbial communities, cataloging the genes prevalent in the gut systems and assessing the extent of HGTs and their effects on the spread of the antibiotic resistance.

The number and size of datasets analyzed as well as the number of reference genomes recovered in this study make this study quite interesting and qualify it to be an important contribution in the field of animal associated microbiome research in general.

Reply: We are grateful for these encouraging comments and for your suggestions to improve our manuscript.

Major takes on this work: -

1- This study is more like data dump than actual analysis paper, it lacks insights, clear conclusions or future recommendations on how to use this massive amount of the data to advance the field.

Reply: We thoroughly revised the manuscript to include insights, conclusions and recommendations.

2- The title and the abstract are bit misleading, they gave me an impression that this study based on datasets created by the group, however it turned to be a re-analysis of publicly available datasets.

Reply: We revised the title and Abstract. We emphasized in the Abstract that the assembled microbial genomes and gene catalog were generated from public chicken gut microbiome samples. (Lines 1-2 and 32-41)

3- The analyses performed on the MAGs were shallow at all levels and I failed to see any conclusive results. The results were more like describing computational tools outputs without spending time or effort to put the results into any context.

Reply: Some of our data were reanalyzed, and necessary interpretations were added throughout the manuscript.

Other points: -

1- Figure 1a, reorder the labels to match the order of the phyla organization on the tree.

Reply: We reordered the labels to match the order of the phyla organization on the tree. In addition, we reconstructed the phylogenetic tree based on the 1,978 MAGs (clustered with 95%

ANI) using PhyloPhlAn (v3.0.60)⁸, which merged all archaeal and bacterial genomes into one tree (Figure 2a). (Lines 99-101 and 265-267)

2- Figure 2b looks like a cartoon than an actual tree with no references to see the position of the taxa, please redo the archaea tree with the addition of references, bootstrap needs to be presented and the phylogenetic trees in newick format should be attached as supplementary documents.

Reply: We reconstructed the phylogenetic tree and merged archaeal and bacterial genomes into one tree (Figure 2a). We also reconstructed the archaeal phylogenetic tree at the strain level using PhyloPhlAn (v3.0.60), and three known archaeal genomes from the three archaeal phyla that we observed were added as references (Supplementary Figure 2). The phylogenetic trees in newick format are included in Supplementary Data 2. (Lines 114-118)

3- Ln 75: Having high diversity doesn't reflect abundance. This statement is not quite accurate. It only reflects a successful niche occupation and probably niche/substrate specializations.

Reply: We revised the description to “Strains of *Firmicutes A* and *Bacteroidota* exhibited the highest diversity, as reflected by the Shannon index (Supplementary Figure 1d), suggesting their contributions to the chicken gut microbiota composition and successful niche occupation and niche/substrate specialization.” (Lines 109-111)

4- Figure S2 need to be normalized to the genome size and data needs to be represented by number of genes/Mbps to reflect the real saccharolytic capabilities of the recovered genomes.

Reply: We have modified Supplementary Figure 1f accordingly. The results showed that microorganisms from the *Bacteroidota* phylum had better saccharolytic capabilities. (Lines 111-114)

5- Ln 107: Some qualitative analysis of the CAZyme classes present is required to get a deeper understanding of the role of CAZymes in niche occupation by the inhabiting microbes.

Reply: We further analyzed the CAZyme classes as suggested, and a new figure, Figure 3, was added. (Lines 154-171)

References

1. Gilroy, R. *et al.* Extensive microbial diversity within the chicken gut microbiome revealed by metagenomics and culture. *PeerJ* **9**, e10941 (2021).

2. Glendinning, L., Stewart, R. D., Pallen, M. J., Watson, K. A. & Watson, M. Assembly of hundreds of novel bacterial genomes from the chicken caecum. *Genome Biol.* **21**, 34 (2020).
3. Huang, P. *et al.* The chicken gut metagenome and the modulatory effects of plant-derived benzylisoquinoline alkaloids. *Microbiome* **6**, 211 (2018).
4. Chen, C. *et al.* Expanded catalog of microbial genes and metagenome-assembled genomes from the pig gut microbiome. *Nat. Commun.* **12**, 1106 (2021).
5. Xie, F. *et al.* An integrated gene catalog and over 10,000 metagenome-assembled genomes from the gastrointestinal microbiome of ruminants. *Microbiome* **9**, 137 (2021).
6. Pasolli, E. *et al.* Extensive unexplored human microbiome diversity revealed by over 150,000 genomes from metagenomes spanning age, geography, and lifestyle. *Cell* **176**, 649-662.e20 (2019).
7. Almeida, A. *et al.* A unified catalog of 204,938 reference genomes from the human gut microbiome. *Nat. Biotechnol.* **39**, 105–114 (2021).
8. Asnicar, F. *et al.* Precise phylogenetic analysis of microbial isolates and genomes from metagenomes using PhyloPhlAn 3.0. *Nat. Commun.* **11**, 2500 (2020).

REVIEWERS' COMMENTS:

Reviewer #1 (Remarks to the Author):

I have no other suggestions.

Reviewer #2 (Remarks to the Author):

The manuscript by Feng et. al has been improved drastically and in my opinion, it can be accepted for publication. However, the minor typography errors must be taken care of in the final version of the accepted article.

Reviewer #3 (Remarks to the Author):

The reviewed version looks a lot better than the original one. I have no further comments.

Responses to the reviewers' comments

Reviewers' comments:

Reviewer #1 (Remarks to the Author):

I have no other suggestions.

Response: Thank you for your time and consideration.

Reviewer #2 (Remarks to the Author):

The manuscript by Feng et. al has been improved drastically and in my opinion, it can be accepted for publication. However, the minor typography errors must be taken care of in the final version of the accepted article.

Response: We have checked the typography errors in the manuscript according to the guides.

Reviewer #3 (Remarks to the Author):

The reviewed version looks a lot better than the original one. I have no further comments.

Response: Thank you for your time and consideration.